# Reconstructing Training Data from Trained Neural Networks

**Niv Haim**[*]
Weizmann Institute of Science
niv.haim@weizmann.ac.il

**Gal Vardi**[*†]
TTI-Chicago and Hebrew University
galvardi@ttic.edu

**Gilad Yehudai**[*]
Weizmann Institute of Science
gilad.yehudai@weizmann.ac.il

**Ohad Shamir**
Weizmann Institute of Science
ohad.shamir@weizmann.ac.il

**Michal Irani**
Weizmann Institute of Science
michal.irani@weizmann.ac.il

Project page: https://giladude1.github.io/reconstruction

## Abstract

Understanding to what extent neural networks memorize training data is an intriguing question with practical and theoretical implications. In this paper we show that in some cases a significant fraction of the training data can in fact be reconstructed from the parameters of a trained neural network classifier. We propose a novel reconstruction scheme that stems from recent theoretical results about the implicit bias in training neural networks with gradient-based methods. To the best of our knowledge, our results are the first to show that reconstructing a large portion of the actual training samples from a trained neural network classifier is generally possible. This has negative implications on privacy, as it can be used as an attack for revealing sensitive training data. We demonstrate our method for binary MLP classifiers on a few standard computer vision datasets.

## 1 Introduction

It is commonly believed that neural networks memorize the training data, even when they are able to generalize well to unseen test data (e.g., [Zhang et al., 2021, Feldman, 2020]). Exploring this memorization phenomenon is of great importance both practically and theoretically. Indeed, it has implications on our understanding of generalization in deep learning, on the hidden representations learnt by neural networks, and on the extent to which they are vulnerable to privacy attacks.

A fundamental question for understanding memorization is:

> *Are the specific training samples encoded in the parameters of a trained classifier?*
> *Can they be recovered from the network parameters?*

In this work, we study this question, and devise a novel scheme which allows us to reconstruct a significant portion of the training data from the parameters of a trained neural network alone, without having any additional information on the data. Thus, we provide a proof-of-concept that the learning

---

[*]Equal contribution, alphabetically ordered
[†]Work done while the author was at the Weizmann Institute of Science

36th Conference on Neural Information Processing Systems (NeurIPS 2022).

(a) Top 24 images reconstructed from a binary classifier trained on 50 CIFAR10 images

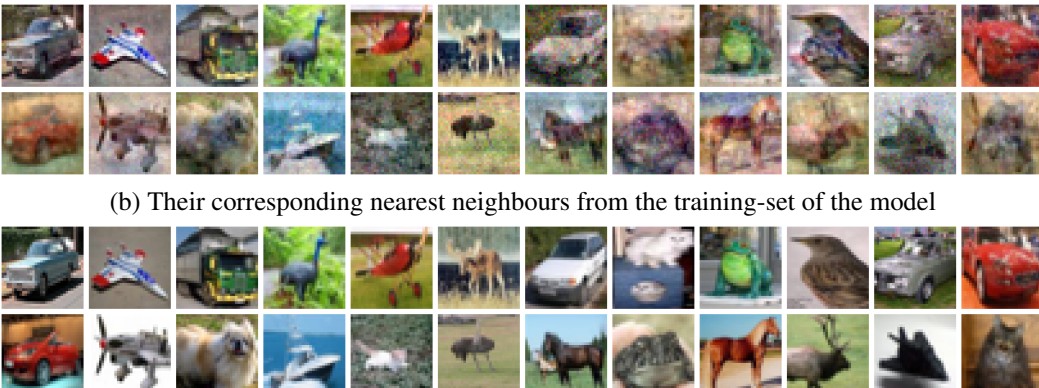

(b) Their corresponding nearest neighbours from the training-set of the model

Figure 1: Reconstruction of training images from a pretrained binary classifier, trained on 50 CIFAR10 images. The two classes are "animals" and "vehicles". We calculate the nearest neighbor using the SSIM metric.

process can sometimes be reversed: That is, instead of learning a model given a training dataset, it is possible to find the training data given a trained model. In Figure 1 we show how our approach reconstructs images from the CIFAR10 dataset, given a simple trained binary classifier.

Many works try to "crack" neural networks by analyzing and visualizing either their learnt parameters or representations [Erhan et al., 2009, Mahendran and Vedaldi, 2015, Olah et al., 2017, 2020]. This is usually done by "inverting" the model, namely finding inputs that are strongly correlated with the model's activations [Mordvintsev et al., 2015, Yin et al., 2020, Fredrikson et al., 2015]. Unsurprisingly, the results are semantically correlated with the training dataset. However, one rarely sees an exact version of a training sample.

Our results have potential negative implications on privacy in deep learning. Our scheme can be viewed as a *training-data reconstruction attack*, since an adversary might recover sensitive training data. For example, if a medical device includes a model trained on sensitive medical records, an adversary might reconstruct this data and thus violate the privacy of the patients. Privacy attacks in deep learning have been widely studied in recent years (cf. Liu et al. [2021]), but as far as we are aware, the known attacks cannot reconstruct portions of the training data from a trained model.

Our approach relies on theoretical results about the implicit bias in training neural networks with gradient-based methods. The implicit bias has been studied extensively in recent years with the motivation of explaining generalization in deep learning (see Section 2). We use results by Lyu and Li [2019], Ji and Telgarsky [2020], which establish that, under some technical assumptions, if we train a neural network with the binary cross entropy loss, its parameters will converge to a stationary point of a certain margin-maximization problem. This result implies that the parameters of the trained network satisfy a set of equations w.r.t. the training dataset. In our approach, given a trained network, we find a dataset that solves this set of equations w.r.t. the trained parameters.

**Our Contributions** We show that large portions of the training samples are encoded in the parameters of a trained classifier. We also provide a practical scheme to decode the training samples, without any assumptions on the data. As far as we know, this is the first work that shows that reconstruction of actual training samples from a trained neural network classifier is possible.

## 2 Related Work

**Understanding and Visualizing what is learnt by Neural Networks.** The most common approach for analysing what is learnt by a neural network is by searching inputs that maximize the class output or the activations of neurons in intermediate layers [Erhan et al., 2009, Olah et al., 2020]. Oftentimes this is done via optimization with respect to the model input. Optimizing without any prior on the input usually results in noise inputs. Therefore, most approaches incorporate priors such as smoothness regularization or the use of pre-trained image generators [Mahendran and Vedaldi, 2015, Yosinski et al., 2015, Mordvintsev et al., 2015, Nguyen et al., 2016a,b, 2017] (see Olah et al. [2017] for a comprehensive summary). Optimization w.r.t. the input may also result in adversarial examples

[Szegedy et al., 2013, Goodfellow et al., 2014]. Recently, [Tsipras et al., 2018, Engstrom et al., 2019] showed that classifiers trained to be robust to adversarial examples tend to learn representations that are more aligned with human vision. This was later utilized by [Santurkar et al., 2019, Mejia et al., 2019] to generate class-conditional images from a trained classifier. While all those approaches indicate that, unsurprisingly, the learnt representations are strongly correlated with the datasets on which the model was trained, none of them demonstrate the reconstruction of exact training samples from the trained models.

**Privacy Attacks in Deep Learning.** Many methods deal with extracting sensitive information from trained models. Perhaps the closest to our approach is *model-inversion* that aims to reconstruct class representatives from the training data of a trained model [Fredrikson et al., 2015, He et al., 2019, Yang et al., 2019, Yin et al., 2020]. It is important to note that the reconstructed images, albeit semantically similar to some input images, are still not actual samples from the training set. Carlini et al. [2021, 2019] demonstrated reconstruction of training data from generative language models. By completing sentences, they reveal sensitive information from the training data. We note that this approach is specific to generative language models, while our approach considers classifiers and is less data specific. *Membership-inference* attacks [Shokri et al., 2017] aim to determine whether a given data point was used to train the model or not. For these methods to work, the adversary must be able to guess a specific input, whereas our approach does not assume such ability. Lastly, avoiding leakage of sensitive information on the training dataset is the motivation behind *differential privacy* in machine learning, which has been extensively studied [Abadi et al., 2016, Dwork et al., 2006, Chaudhuri et al., 2011]. For an elaborated discussion on the relation of these approaches to ours see Appendix A.

**Implicit Bias.** In overparameterized neural networks one might expect overfitting to occur, but it seems that gradient-based methods are biased towards networks that generalize well [Zhang et al., 2021, Neyshabur et al., 2017]. Mathematically characterizing this implicit bias is a major problem in the theory of deep learning. Our approach is based on a characterization of the implicit bias of gradient flow in homogeneous neural networks due to Lyu and Li [2019] and Ji and Telgarsky [2020] (see Section 3 for details). The implicit bias of gradient-based methods in neural networks was extensively studied in recent years both for classification tasks (e.g., Soudry et al. [2018], Gunasekar et al. [2018c], Ji and Telgarsky [2018], Nacson et al. [2019], Vardi et al. [2021], Chizat and Bach [2020], Gunasekar et al. [2018a], Moroshko et al. [2020]) and regression tasks (e.g., Gunasekar et al. [2018b], Arora et al. [2019], Azulay et al. [2021], Yun et al. [2020], Woodworth et al. [2020], Razin and Cohen [2020], Li et al. [2020], Vardi and Shamir [2021], Timor et al. [2022]). See Vardi [2022] for a survey.

## 3 Background and Reconstruction Scheme

In this section we present our training data reconstruction scheme, as well as provide a brief overview on the theoretical results about implicit bias, which motivate our approach.

### 3.1 On the Implicit Bias of Neural Networks

Let $S = \{(\mathbf{x}_i, y_i)\}_{i=1}^n \subseteq \mathbb{R}^d \times \{-1, 1\}$ be a binary classification training dataset. Let $\Phi(\boldsymbol{\theta}; \cdot) : \mathbb{R}^d \to \mathbb{R}$ be a neural network parameterized by $\boldsymbol{\theta} \in \mathbb{R}^p$. For a loss function $\ell : \mathbb{R} \to \mathbb{R}$ the empirical loss of $\Phi(\boldsymbol{\theta}; \cdot)$ on the dataset $S$ is $\mathcal{L}(\boldsymbol{\theta}) := \sum_{i=1}^n \ell(y_i \Phi(\boldsymbol{\theta}; \mathbf{x}_i))$. We focus on the *logistic loss* (a.k.a. *binary cross entropy*), namely, $\ell(q) = \log(1 + e^{-q})$.

Our approach is based on Theorem 3.1 below, which holds for *gradient flow* (i.e., gradient descent with an infinitesimally small step size). Before stating the theorem, we need the following definitions: (1) We say that gradient flow *converges in direction* to $\tilde{\boldsymbol{\theta}}$ if $\lim_{t \to \infty} \frac{\boldsymbol{\theta}(t)}{\|\boldsymbol{\theta}(t)\|} = \frac{\tilde{\boldsymbol{\theta}}}{\|\tilde{\boldsymbol{\theta}}\|}$, where $\boldsymbol{\theta}(t)$ is the parameter vector at time $t$; (2) We say that a network $\Phi$ is *homogeneous* w.r.t. the parameters $\boldsymbol{\theta}$ if there exists $L > 0$ such that for every $\alpha > 0$ and $\boldsymbol{\theta}, \mathbf{x}$ we have $\Phi(\alpha\boldsymbol{\theta}; \mathbf{x}) = \alpha^L \Phi(\boldsymbol{\theta}; \mathbf{x})$. Thus, scaling the parameters by any factor $\alpha > 0$ scales the outputs by $\alpha^L$. We note that essentially any fully-connected or convolutional neural network with ReLU activations is homogeneous w.r.t. the parameters $\boldsymbol{\theta}$ if it does not have any skip-connections (i.e., residual connections) or bias terms, except possibly for the first layer.

**Theorem 3.1 (Paraphrased from Lyu and Li [2019], Ji and Telgarsky [2020])** *Let $\Phi(\boldsymbol{\theta}; \cdot)$ be a homogeneous ReLU neural network. Consider minimizing the logistic loss over a binary classification dataset $\{(\mathbf{x}_i, y_i)\}_{i=1}^n$ using gradient flow. Assume that there exists time $t_0$ such that $\mathcal{L}(\boldsymbol{\theta}(t_0)) < 1^{\ddagger}$. Then, gradient flow converges in direction to a first order stationary point (KKT point) of the following maximum-margin problem:*

$$\min_{\boldsymbol{\theta}'} \frac{1}{2} \left\| \boldsymbol{\theta}' \right\|^2 \quad s.t. \quad \forall i \in [n] \;\; y_i \Phi(\boldsymbol{\theta}'; \mathbf{x}_i) \geq 1 \; . \tag{1}$$

*Moreover, $\mathcal{L}(\boldsymbol{\theta}(t)) \to 0$ as $t \to \infty$.*

The above theorem guarantees directional convergence to a first order stationary point (of the optimization problem (1)), which is also called *Karush–Kuhn–Tucker point*, or *KKT point* for short. The KKT approach allows inequality constraints, and is a generalization of the method of *Lagrange multipliers*, which allows only equality constraints.

The great virtue of Theorem 3.1 is that it characterizes the *implicit bias* of gradient flow with the logistic loss for homogeneous networks. Namely, even though there are many possible directions of $\frac{\boldsymbol{\theta}}{\|\boldsymbol{\theta}\|}$ that classify the dataset correctly, gradient flow converges only to directions that are KKT points of Problem (1). In particular, if the trajectory $\boldsymbol{\theta}(t)$ of gradient flow under the regime of Theorem 3.1 converges in direction to a KKT point $\tilde{\boldsymbol{\theta}}$, then we have the following: There exist $\lambda_1, \ldots, \lambda_n \in \mathbb{R}$ such that

$$\tilde{\boldsymbol{\theta}} = \sum_{i=1}^n \lambda_i y_i \nabla_{\boldsymbol{\theta}} \Phi(\tilde{\boldsymbol{\theta}}; \mathbf{x}_i) \qquad \text{(stationarity)} \tag{2}$$

$$\forall i \in [n], \;\; y_i \Phi(\tilde{\boldsymbol{\theta}}; \mathbf{x}_i) \geq 1 \qquad \text{(primal feasibility)} \tag{3}$$

$$\lambda_1, \ldots, \lambda_n \geq 0 \qquad \text{(dual feasibility)} \tag{4}$$

$$\forall i \in [n], \;\; \lambda_i = 0 \text{ if } y_i \Phi(\tilde{\boldsymbol{\theta}}; \mathbf{x}_i) \neq 1 \qquad \text{(complementary slackness)} \tag{5}$$

Our main insight is based on Eq. (2), which implies that the parameters $\tilde{\boldsymbol{\theta}}$ are a linear combinations of the derivatives of the network at the training data points. We say that a data point $\mathbf{x}_i$ is *on the margin* if $y_i \Phi(\tilde{\boldsymbol{\theta}}; \mathbf{x}_i) = 1$ (i.e. $|\Phi(\tilde{\boldsymbol{\theta}}; \mathbf{x}_i)| = 1$). Note that Eq. (5) implies that only samples which are on the margin affect Eq. (2), since samples not on the margin have a coefficient $\lambda_i = 0$.

## 3.2 Dataset Reconstruction

Suppose we are given a trained neural network with parameters $\boldsymbol{\theta}$, and our goal is to reconstruct the dataset that the network was trained on. Although Theorem 3.1 holds asymptotically as the time $t$ tends to infinity, it suggests that also after training for a finite number of iterations the parameters of the network might approximately satisfy Eq. (2), and the coefficients $\lambda_i$ satisfy Eq. (4). Since $n$ is unknown (and so is the number of samples on the margin) we set $m \geq 2n$ which represents the number of samples we want to reconstruct (thus, we only need to upper bound $n$), and fix $y_i = 1$ for $i = 1, \ldots, m/2$ and $y_i = -1$ for $i = m/2 + 1, \ldots, m$. We define the following losses:

$$L_{\text{stationary}}(\mathbf{x}_1, \ldots, \mathbf{x}_m, \lambda_1, \ldots, \lambda_m) = \left\| \boldsymbol{\theta} - \sum_{i=1}^m \lambda_i y_i \nabla_{\boldsymbol{\theta}} \Phi(\boldsymbol{\theta}; \mathbf{x}_i) \right\|_2^2 \tag{6}$$

$$L_\lambda(\lambda_1, \ldots, \lambda_m) = \sum_{i=1}^m \max\{-\lambda_i, 0\} \tag{7}$$

Note that the unknown parameters are the $\mathbf{x}_i$'s and $\lambda_i$'s, and that $\boldsymbol{\theta}$ and the $y_i$'s are given. The loss $L_{\text{stationary}}$ represents the stationarity condition that the parameters of the network satisfy, and $L_\lambda$ represents the dual feasibility condition. We additionally define $L_{\text{prior}}$ which represents some prior knowledge we might have about the dataset. For example, if we know that the dataset contains images, prior knowledge would be that each input coordinate (i.e. each pixel) is between $0$ and $1$. Given no prior knowledge on the data, we can define $L_{\text{prior}} \equiv 0$. Finally, we define the reconstruction loss as:

$$L_{\text{reconstruct}}(\{\mathbf{x}_i\}_{i=1}^m, \{\lambda_i\}_{i=1}^m) = \alpha_1 L_{\text{stationary}} + \alpha_2 L_\lambda + \alpha_3 L_{\text{prior}} \tag{8}$$

---

[‡]This ensures that $\ell(y_i \Phi(\boldsymbol{\theta}(t_0); \mathbf{x}_i)) < 1$ for all $i$, i.e. at some time $\Phi$ classifies every sample correctly.

where $\alpha_1, \alpha_2, \alpha_3 \in \mathbb{R}$ are tunable hyperparameters of the different losses. To reconstruct the dataset, we can use any nonconvex optimization method (e.g. SGD) to find the $\mathbf{x}_1, \ldots, \mathbf{x}_m, \lambda_1, \ldots, \lambda_m$ which minimize Eq. (8). We note that the $\lambda_i$'s are not part of the training data, but finding them is necessary in order to solve this optimization problem. Finally, we emphasize that there are many other possible options to formulate the KKT conditions Eq. (2)-(5) as an unconstrained optimization problem. However, this simple choice seemed to work quite well in practice.

We note that if there exist $\{\mathbf{x}_i\}_{i=1}^n$ and $\{\lambda_i\}_{i=1}^n$ which satisfy the KKT conditions, then there are $\{\mathbf{x}_i\}_{i=1}^m$ and $\{\lambda_i\}_{i=1}^m$ which achieve zero loss in Eq. (8). Indeed, such a solution can be obtained by adding to $\{\mathbf{x}_i\}_{i=1}^n$ additional points $\mathbf{x}_j$ with $\lambda_j = 0$, or by duplicating some points in $\{\mathbf{x}_i\}_{i=1}^n$ and modifying the $\lambda$'s accordingly. Also, note that since we choose $m \geq 2n$, then we set at least $n$ labels $y_i$ to 1 and at least $n$ labels to $-1$. Hence, there is a solution to Eq. (8) even though we do not know the real distribution of labels in the actual training data.

We cannot simply use Eq. (3) and (5) in our reconstruction scheme, because they contain the constant "1" which corresponds to the margin (i.e., $\min_i |\Phi(\tilde{\boldsymbol{\theta}}; \mathbf{x}_i)|$). Namely, we only converge *in direction* to a point $\tilde{\boldsymbol{\theta}}$ that attains margin 1, but in practice we approach some point $\boldsymbol{\theta}$ which attains an unknown margin $\gamma$ (i.e., $\min_i |\Phi(\boldsymbol{\theta}; \mathbf{x}_i)| = \gamma$), and we do not know in advance how to normalize it to attain a margin of exactly 1. On the other hand, Eq. (2) and (4) hold not only for $\tilde{\boldsymbol{\theta}}$ but also for any $\boldsymbol{\theta}$ that points at the direction of $\tilde{\boldsymbol{\theta}}$, and therefore in our loss in Eq. (8) we rely only on these conditions.

Intuitively, a reason to believe that there is enough information in Eq. (2) to reconstruct the data, is the following observation: Eq. (2) represents a set of $p$ equations with $O(nd)$ unknown variables, where $p$ is the number of parameters in the network. In practice, neural networks are often highly overparameterized (i.e., $p > nd$), suggesting more equations than variables.

Finally, since by Eq. (5) we have $\lambda_i = 0$ for every $\mathbf{x}_i$ that is not on the margin, then Eq. (2) implies that $\tilde{\boldsymbol{\theta}}$ is determined only by the gradients w.r.t. the data points that are on the margin. Hence, we can only expect to reconstruct training samples that are on the margin (see also Subsection 5.3).

## 4    A Simple Experiment in Two Dimensions

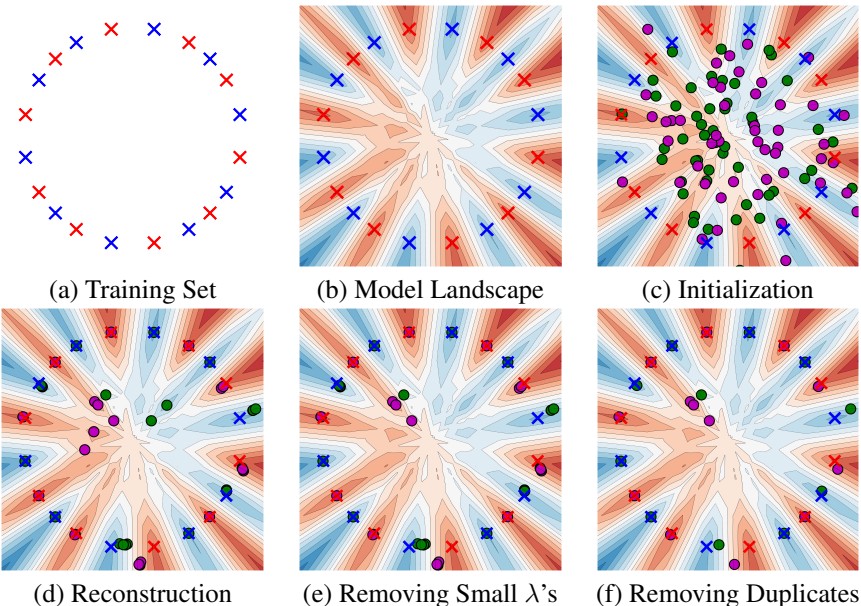

|            |              |            |
|:----------:|:------------:|:----------:|
| (a) Training Set | (b) Model Landscape | (c) Initialization |
| (d) Reconstruction | (e) Removing Small $\lambda$'s | (f) Removing Duplicates |

Figure 2: Exemplifying our reconstruction scheme on a simple 2D dataset (see text for explanation).

In this section we exemplify our dataset reconstruction scheme on a toy example of 2-dimensional data, i.e. we consider $(\mathbf{x}, y) \in \mathbb{R}^2 \times \{\pm 1\}$. We set $n = 20$ training samples on the unit circle, with alternating labels. For a visualization of the dataset see Figure 2a, blue and red "$\times$" represent the two classes. We trained a 3-layer model with 1000 neurons in each layer on this dataset. The model learns

to correctly classify the training set. In Figure 2b, we visualize the output of the model as a function of its input. Blue and red regions correspond to smaller and larger outputs of the model, respectively.

We now demonstrate our reconstruction scheme. We first randomly initialize $m = 100$ points in $\mathbb{R}^2$, and assign 50 points to each class. This is depicted in Figure 2c, where green points correspond to the blue class, and magenta points correspond to the red class. Next, we optimize the loss in Eq. (8), with $L_{\text{prior}} \equiv 0$. The results of our reconstruction scheme are in Figure 2d. Note that our approach reconstructed all the input samples, up to some noise.

To further improve our reconstruction results, we remove some of the extra points which did not converge to a training sample. In Figure 2e we removed points $\mathbf{x}_i$ with corresponding $\lambda_i < 5$. According to Eq. (2), points with $\lambda_i = 0$ should not affect the parameters, hence their corresponding $\mathbf{x}_i$ can take any value. In practice, it is sufficient to remove points with a small enough corresponding $\lambda_i$. Finally, to remove duplicates, we greedily remove points which are very close to other points. That is, we randomly order the points, and iteratively remove points that are at distance $< 0.03$ from another point. The final reconstruction result is depicted in Figure 2f.

## 5   Results

Top 45 images reconstructed from a model trained on CIFAR10 (rows $1, 3, 5$), and their corresponding nearest-neighbors from the training-set of the model (rows $2, 4, 6$)

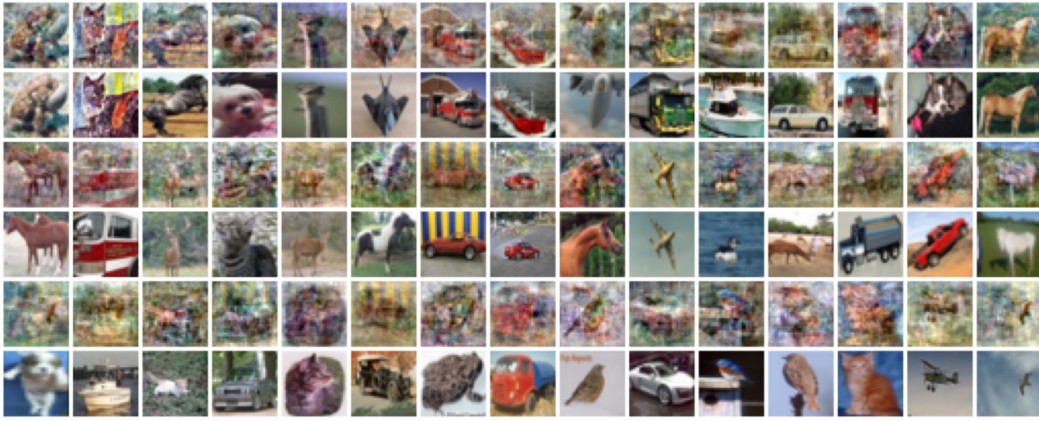

Top 45 images reconstructed from a model trained on MNIST (rows $1, 3, 5$), and their corresponding nearest-neighbors from the training-set of the model (rows $2, 4, 6$)

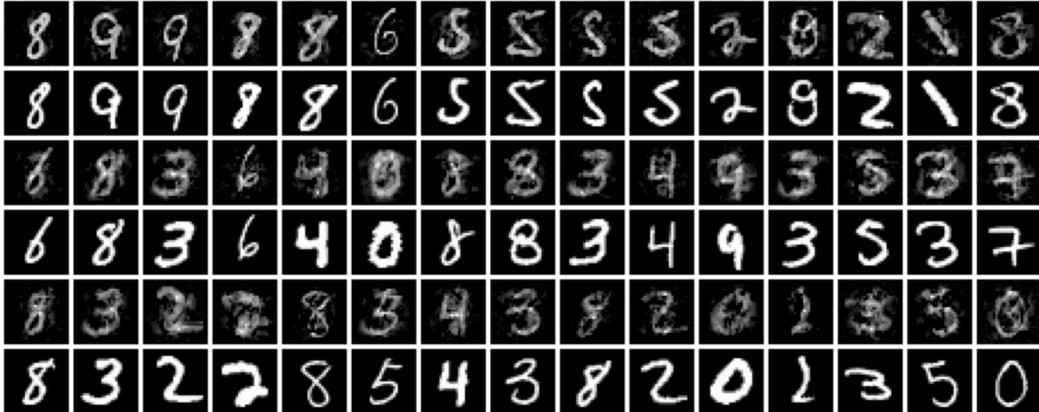

Figure 3: Reconstructing training samples from two binary classifiers – one trained on 500 images with labels animals/vehicles (CIFAR), and the other trained on 500 odd/even digit images (MNIST). Train errors are zero, test accuracies are $88.0\%/77.6\%$ for MNIST/CIFAR

## 5.1 Experimental Setup

**Datasets.** We conduct experiments on binary classification tasks where images are taken from the MNIST [LeCun et al., 2010] and CIFAR10 [Krizhevsky et al., 2009] datasets and the labels are set to odd vs. even digits (MNIST), and vehicles vs. animals[§] (CIFAR10). We make sure that the class distribution in the training and test sets is balanced, and normalize the train and test sets by reducing the mean of the training set from both.

**Training.** We consider MLP architectures. Unless stated otherwise, our models comprise of three fully-connected layers with dimensions $d$-1000-1000-1 (where $d$ is the dimension of the input) with ReLU activations. Biases are set to zero except for the first layer, to line up with the theoretical assumption of homogeneous models in Section 3. The parameters are initialized using standard Kaiming He initialization [He et al., 2015] except for the weights of the first layer that are initialized to a Gaussian distribution with standard deviation $10^{-4}$ (see discussion in Subsection 5.2). We train our models using full batch gradient descent for $10^6$ epochs with a learning rate of 0.01. All models achieve zero training error (i.e., all the train samples are labeled correctly), and a training loss $< 10^{-6}$. To compute the test accuracy, we use the original test sets of MNIST/CIFAR10 with 10000/8000 images respectively, and labeled accordingly.

## 5.2 Training Set Reconstruction

We minimize the loss defined in Eq. (8) with $\alpha_1 = 1$, $\alpha_2 = 5$, $\alpha_3 = 1$. We initialize $\mathbf{x}_i \sim \mathcal{N}(0, \sigma_x I)$, where $\sigma_x$ is a hyperparameter, and $\lambda_i \sim \mathcal{U}[0, 1]$. We set the number of reconstructed samples to $m = 2n$ (where $n$ is the size of the original training set). Note that our loss contains the derivative of ReLU Eq. (6). This derivative is a step function, containing only flat regions which are hard to optimize. We replace the derivative of the ReLU layer (backward function) with a sigmoid, which is the derivative of softplus (a smooth version of ReLU). We use the fact that our inputs are images to penalize values outside the range $[-1, 1]$. To this end we set $L_{\text{prior}}(z) = \max\{z - 1, 0\} + \max\{-z - 1, 0\}$ for each pixel $z$, and average over all dimensions (pixels) in $\mathbf{x}_i$. We optimize our loss for $100,000$ iterations using an SGD optimizer with momentum 0.9. We conduct a total of 100 runs using a random grid search on the hyperparameters (e.g. learning rate, $\sigma_x$. See Appendix B for full details). This results in $100m$ "reconstructed" inputs.

While some $\mathbf{x}_i$ end up converging to a training sample, some end as noise (similar phenomenon can be observed in 2D in Figure 2d). To identify the reconstructions that are most similar to a training image we use the SSIM metric [Wang et al., 2004].

In Figure 3 we show the best reconstruction results (in terms of SSIM) for models trained on $n$=500 samples from MNIST/CIFAR10 datasets (with test accuracy 88.0%/77.6% resp.). Note that the reconstructed images are very similar to the real input data, although a bit noisy. The source of this noise is not entirely clear. Possible reasons may be the complexity of the optimization problem, or the possibility that the trained model has not fully converged to the KKT point of Problem (1).

We observed that small initializations significantly improve the quality of the reconstructed samples. We conjecture that small initialization causes faster convergence to the direction of the KKT point. This is also theoretically implied in Moroshko et al. [2020] (for certain linear models). Similarly, training for more epochs also improves the quality of the reconstruction. In Appendix C we show results for reconstructions from networks trained with standard initialization or trained for much fewer epochs. During the training phase, we used full batch gradient descent, to remain as much aligned to the theoretical setting. In Appendix C we show that our approach can reconstruct training data also from models trained with mini-batch SGD.

## 5.3 Practice vs. Theory

In this section we analyze some relations between our experimental results to the theory laid down in Section 3. Given a trained model and its reconstructed samples, we match each training sample to its best reconstruction (in terms of SSIM score). We then plot this SSIM score against $\Phi(\boldsymbol{\theta}; \mathbf{x})$ (the value of the model's output on this training sample) – for all training samples. In Figure 4 each cell shows such plot for a given model. The top row shows models trained on the same architecture with

---

[§]Automobile, Truck, Airplane, Ship vs. Bird, Horse, Cat, Dog, Deer, Frog.

Models with the same architecture (1000-1000) trained on different number of training samples ($n$)

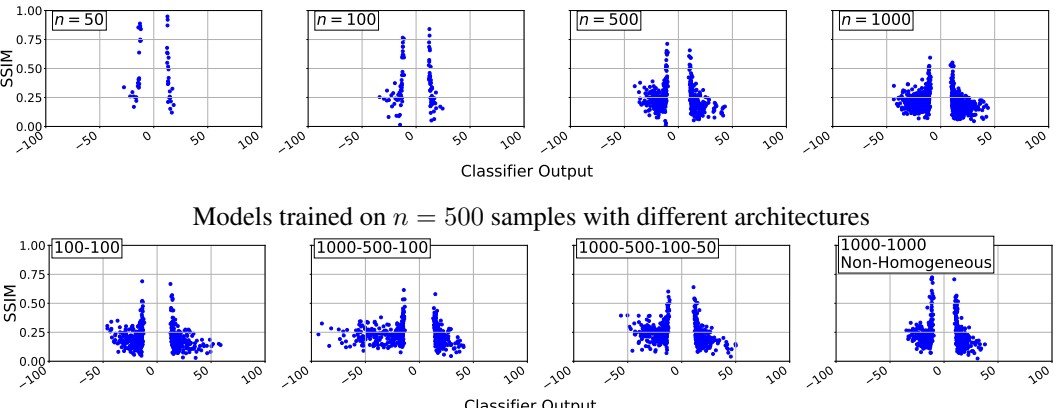

Models trained on $n = 500$ samples with different architectures

Figure 4: Each point represents a training sample. The y-axis is the highest SSIM score achieved by a reconstruction of this sample, the x-axis is the output of the model. **Top:** The effect of training the same model on different number of training samples ($n$). **Bottom:** The effect of training models with different architectures (on $n = 500$ training samples). The right-most plot shows a 3-layer non-homogeneous MLP (with bias terms in all hidden layers). See discussion in Section 5.3.

different number of training samples ($n$), where in the bottom row we show the results for models trained on $n = 500$ training samples, with different architectures (all results are on CIFAR10).

Recall that we do not expect to reconstruct samples that are far from the margin (Subsection 3.2). It is evident from Figure 4 that good reconstructions (e.g., SSIM$> 0.4$) are obtained for samples that lie on the margin, as expected from theory. The plots indicate that increasing training size makes reconstruction more difficult. Lastly, as seen from the rightmost plot in the bottom row, we manage to get high-quality reconstructions from a non-homogeneous model (trained with biases in all hidden layers). This indicates that our approach may work beyond the theoretical limitations of Theorem 3.1.

### 5.4  Comparison to other Reconstruction Schemes

**Model Inversion.**   Given a trained model $\Phi(\boldsymbol{\theta}; \cdot)$, we search for $\mathbf{x}$ which maximizes or minimizes $\Phi(\boldsymbol{\theta}; \mathbf{x})$, corresponding to positive or negative labels. We initialize $\mathbf{x} \sim \mathcal{N}(0, \sigma I)$ for several values of $\sigma$ and optimize w.r.t. the model output (see Appendix B for the choice of hyperparameters). In Figure 5a (left) it is apparent that in our two-dimensional experiment, model inversion successfully reconstructed 7 training samples, which indeed lie on a local minimum or maximum. However, note that our scheme reconstructs all 20 samples (Figure 2). In high dimensions, namely, in MNIST and CIFAR, while our scheme can reconstruct a large portion of the training set (Figure 3a), model inversion converges to noisy/blurry class representatives that correspond to high/low output values (Figure 5a, right). Such results are typical with model inversion since not all class members from the training set are visually similar (see discussions in Shokri et al. [2017], Melis et al. [2019]).

**Weights Visualization.**   The weights of the first fully-connected layer have the same dimension as that of the input. One may wonder whether training samples are directly encoded there. In Figure 5b we show the weights that are most similar (SSIM) to a training sample, or all of them in the 2D case. As seen in the 2D case, most weights are in the general direction of a training sample, however the scale is unknown without prior knowledge on the data. For images (MNIST/CIFAR10), not more than 3 or 4 of the weights have resemblance to training samples, while our scheme manages to reconstruct dozens of samples. See Appendix B and C for details and all 1000 weights of the models.

## 6  Discussion and Conclusion

Even though our results are shown for relatively small-scale models, they are the first to show that the parameters of trained networks may contain enough information to fully reconstruct training samples,

(a) Model Inversion

2D                CIFAR10

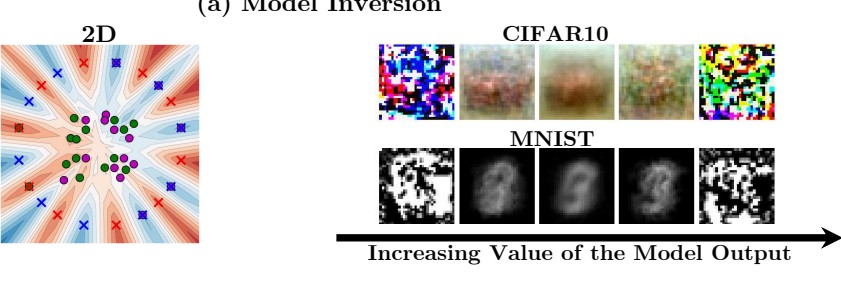

MNIST

**Increasing Value of the Model Output**

(b) Weights of the first Fully-Connected Layer

2D

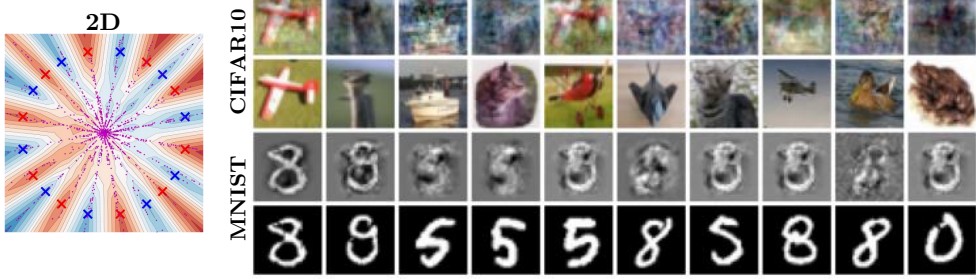

Figure 5: Comparison to other reconstruction schemes. **Top:** Model inversion on the 2D experiment (left), on CIFAR10 (top right) and MNIST (bottom right). The CIFAR and MNIST images are ordered by the value of their output from left (smallest) to right (largest). **Bottom:** Weights of the first (fully-connected) layer for the 2D experiment (left), CIFAR10 (top right) and MNIST (bottom right). The weights for the 2D experiment are the small purple dots. For the CIFAR and MNIST experiments we show the 10 weights with the highest SSIM score.

and the first to *reconstruct a substantial amount of training samples*. Moreover, the theoretical basis of the implicit bias in neural networks provides an analytic explanation to this phenomenon.

Solving our optimization problem for convolutional neural networks turned out to be more challenging and is therefore a subject of future research. We note that the theoretical results that we rely on (i.e., Theorem 3.1) also covers convolutional neural networks. We believe that the homogeneity restriction might be relaxed, and showed reconstructions also from a non-homogeneous model (Figure 4, bottom-rightmost). We also believe that our method may be extended to multi-class classifiers using an extension of Theorem 3.1. Finally, showing reconstructions on larger models and datasets, or on tabular or textual data are interesting future directions.

On the theoretical side, it is not entirely clear why our optimization problem in Eq. (8) converges to actual training samples, even though there is no guarantee that the solution is unique, especially when using no prior (other than simple bounding to $[-1, 1]$). As a final note, our work brings up the question: *are samples on margin the only ones that can be recovered from a trained classifier?* or there exist better reconstruction schemes to reconstruct even more training samples from a trained neural network.

### Acknowledgements

This project received funding from the European Research Council (ERC) under the European Union's Horizon 2020 research and innovation programme (grant agreement No 788535), and ERC grant 754705, and from the D. Dan and Betty Kahn Foundation, and was supported by the Carolito Stiftung.

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
