# Appendix

## Table of Contents

## A   More Details on Privacy Attacks in Deep Learning

Below we discuss several privacy attacks that have been extensively studied in recent years (see Liu et al. [2021], Jegorova et al. [2021] for surveys).

**Membership Inference.**   In *membership-inference attacks* [Shokri et al., 2017, Long et al., 2018, Salem et al., 2018, Yeom et al., 2018, Song and Mittal, 2021] the adversary determines whether a given data point was used to train the model or not. For example, if the model was trained on records of patients with a certain disease, the adversary might learn that an individual's record appeared in the training set and thus infer that the owner of the record has the disease with high chance. Note that membership inference attacks are significantly different from our attack, as the adversary must choose a specific data point. E.g., if the inputs are images, then the adversary must be able to guess a specific image.

**Model Extraction.**   In *model-extraction attacks* [Tramèr et al., 2016, Oh et al., 2019, Wang and Gong, 2018, Carlini et al., 2020b, Jagielski et al., 2020, Milli et al., 2019, Rolnick and Kording, 2020, Chen et al., 2021] the adversary aims to steal the trained model functionality. In this attack, the adversary only has black-box access with no prior knowledge of the model parameters or training data, and the outcome of the attack is a model that is approximately the same as the target model. It was shown that in certain cases the adversary can reconstruct the exact parameters of the target model. We note that such attacks might be combined with our attack in order to allow extraction of the training dataset in a black-box setting. Namely, in the first stage the model is extracted using model-extraction attacks, and in the second stage the training dataset is reconstructed using our attack.

**Model Inversion.**   *Model-inversion attacks* [Fredrikson et al., 2015] are perhaps the closest to our attack, as they consider reconstruction of input data given a trained model. These attacks aim to infer class features or construct class representatives, given that the adversary has some access (either black-box or white-box) to a model.

Fredrikson et al. [2015] showed that a face-recognition model can be used to reconstruct images of a certain person. This is done by using gradient descent for obtaining an input that maximizes the output probability that the face-recognition model assigns to a specific class. Thus, if a class contains only images of a certain individual, then by maximizing the output probability for this class we obtain an image that might be visually similar to an image of that person. It is important to note that the reconstructed image is not an actual example from the training set. Namely, it is an image that contains features which the classifier identifies with the class, and hence it might be visually similar to any image of the individual (including images from the training set). If the class members are not

all visually similar (which is generally the case), then the results of model inversion do not look like the training data (see discussions in Shokri et al. [2017] and Melis et al. [2019]). For example, if this approach is applied to the CIFAR-10 dataset, it results in images which are not human-recognizable [Shokri et al., 2017]. In Zhang et al. [2020], the authors leverage partial public information to learn a distributional prior via generative adversarial networks (GANs) and use it to guide the inversion process. That is, they generate images where the target model outputs a high probability for the considered class (as in Fredrikson et al. [2015]), but also encourage realistic images using GAN. We emphasize that from the reasons discussed above, this method does not reconstruct any specific training data point. Another approach for model inversion is training a model that acts as an inverse of the target model [Yang et al., 2019]. Thus, the inverse model takes the predicted confidence vectors of the target model as input, and outputs reconstructed data. A recent paper Balle et al. [2022] shows a reconstruction attack where the attacker has information about all the data samples except for one. On the theoretical side, Brown et al. [2021] prove that in certain settings, models memorize information about training examples, and show reconstruction attacks on some synthetic datasets.

Model inversion and information leakage in *collaborative deep learning* was studied in, e.g., He et al. [2019], Melis et al. [2019], Hitaj et al. [2017], Zhu et al. [2019], Yin et al. [2021], Huang et al. [2021]. Extraction of training data from language models was studied in Carlini et al. [2021, 2019], where they use the ability of language models to complete a given sentence in order to reveal sensitive information from the training data. We note that this attack is specific to language models, which are generative models, while our approach considers classifiers and is less specific.

**Defences against Training Data Reconstruction.** Avoiding leakage of sensitive information on the training dataset is the motivation behind *differential privacy* in machine learning, which has been extensively studied in recent years [Abadi et al., 2016, Dwork et al., 2006, Chaudhuri et al., 2011]. This approach allows provable guarantees on privacy, but it typically comes with high cost in accuracy. Other approaches for protecting the privacy of the training set, which do not allow such provable guarantees, have also been suggested (e.g., Huang et al. [2020], Carlini et al. [2020a]).

## B  Implementation Details

### B.1  Hardware, Software and Running Time

A typical reconstruction runs for about 30 minutes on a GPU Tesla V-100 32GB, for reconstructing $m = 1000$ samples from a model with architecture $d$-1000-1000-1, and for $100,000$ epochs (running times slightly differ with the number of samples $m$, number of epochs and the size of the model, but it still takes about this time to run). Our code is implemented in PYTORCH [Paszke et al., 2019]. We will release the code.

### B.2  Hyperparameters

Our reconstruction scheme has 4 hyperparameters. Already discussed in the paper are the learning rate and $\sigma_{\mathbf{x}}$ (discussed in Subsection 5.2). In Subsection 5.2 we discuss the modification in the derivative of a ReLU layer $y = \max\{0, x\}$. The backward function of a ReLU layer works as follows: given the "gradient from above" $\frac{\partial L}{\partial y}$, the backward gradient is $\frac{\partial L}{\partial y} \cdot \mathbb{I}\{\mathbf{x} > 0\}$. Our modification to the backward gradient is $\frac{\partial L}{\partial y} \cdot \sigma(\alpha \mathbf{x})$, where $\sigma(z) = \frac{1}{1+e^{-z}}$ and $\alpha$ is a hyperparameter. As noted in the paper, this derivative is essentially the derivative of a SoftReLU, where the derivative is the same as ReLU for $\alpha \to \infty$ and is the derivative of the identity function for $\alpha \to 0$. Note that this is done only in the backward function, while the forward function remains that of a ReLU function. We also add an extra hyperparameter $\lambda_{\min}$ to our $L_\lambda$ loss from Eq. (7):

$$L_\lambda(\lambda_1, \ldots, \lambda_m) = \sum_{i=1}^{m} \max\{-\lambda_i + \lambda_{\min}, 0\}$$

The intuition behind is to encourage as many samples to lie on a margin, and thus try and reconstruct some sample from the training set.

To sum it all, the hyperparameters of our reconstruction scheme are:

1. Reconstruction learning rate
2. $\sigma_{\mathbf{x}}$, the initial scale of $x_i$ initialization

3. $\alpha$, of the derivative of the modified ReLU

4. $\lambda_{\min}$

To find the set of hyperparamerers we used Weights&Biases [Biewald, 2020] using a random grid search where the parameters are sampled from the following distributions:

- Learning rate, log-uniform in $[10^{-5}, 1]$

- $\sigma_{\mathbf{x}}$, log-uniform in $[10^{-6}, 1]$

- ReLU derivative $\alpha$, uniform in $[10, 500]$

- $\lambda_{\min}$, log-uniform in $[10^{-4}, 1]$

When searching for hyperparameters for the model inversion results in Subsection 5.4 we use the following:

- Learning rate, log-uniform in $[10^{-6}, 1]$

- $\sigma_{\mathbf{x}}$, log-uniform in $[10^{-7}, 1]$

### B.3   Post-Processing of Reconstructed Samples

After the reconstruction run ends we want to match the reconstructed samples to samples from the training set. This is done in the following manner:

1. **Scaling.** Each reconstructed sample is stretched to fit into the range $[0, 1]$ (by linear transformation of its minimal/maximal values).

2. **Searching Nearest Neighbours.** For each training sample from the training set we compute the distance to all reconstructed outputs using NCC [Lewis, 1995].

3. **Voting.** For each training sample we compute the mean of all the closest nearest neighbours (all reconstructed samples with NCC score largest than 0.9 of the distance to the closest nearest neighbour). Now we have pairs of trainig-sample and its reconstruction.

4. **Sorting.** For each pair we compute its SSIM [Wang et al., 2004], and sort the results by descending order.

## C   Supplementary Results

### C.1   Results for Models in Figure 4

In this subsection we provide more details and experiments on each model presented in Figure 4. In Table 1 we show the train loss, test error and test loss of each model from Figure 4. All the models achieved a train accuracy of $100\%$. We note that adding more training samples improves the test accuracy, while adding more layers keeps the test accuracy approximately the same. In Figures 6-11 we show the best $45$ extracted images (sorted by SSIM score) for the models presented in Figure 4. The reconstructions for the $50$ and $500$ samples with a $d$-1000-1000-1 architecture is presented in Figure 1 and Figure 3 (top) respectively.

### C.2   All Comparisons for Subsection 5.4

In this subsection we provide more detailed results on the comparison to other methods as presented in Subsection 5.4. In Figure 12 and Figure 13 we provide more results from the model inversion attack on models trained on CIFAR10 and MNIST respectively. These are the same models from Figure 5 (a). In this attack, we either minimize or maximize the model's output w.r.t. a randomly initialized input. In this experiment, half of the initializations were maximized and the other half is minimized. The images are ordered by output of the model, in an increasing order. The results indicate that the model inversion attack mostly converge to similar reconstructions, even with many different initializations and different hyperparameters. Also, these reconstruction are mostly blurry, and probably represent the averages of each class.

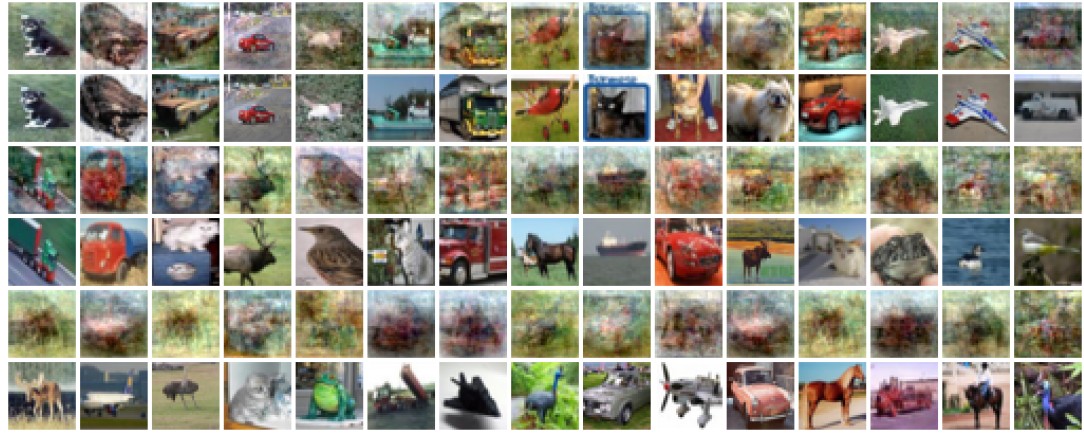

Figure 6: **Architecture:** $d$-1000-1000-1, **Samples:** 100
Odd rows (1,3,5) are reconstructions, even rows (2,4,6) are the original data.

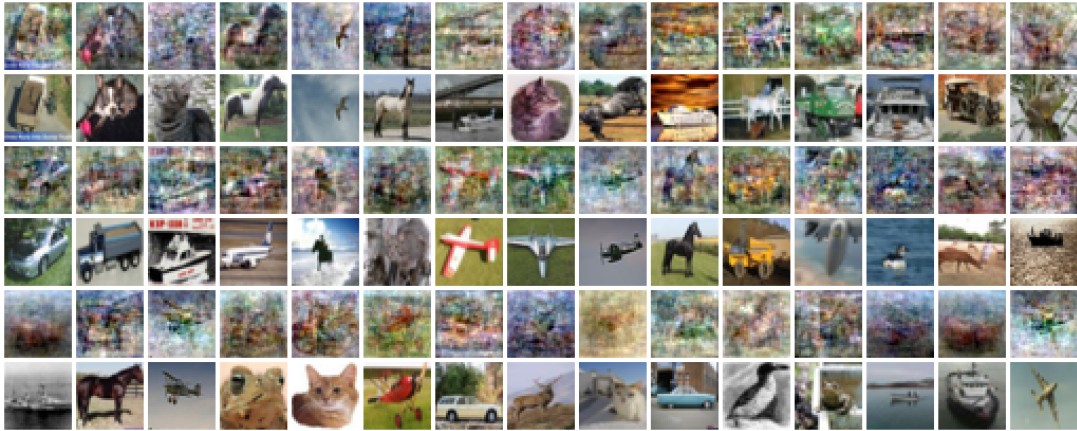

Figure 7: **Architecture:** $d$-1000-1000-1, **Samples:** 1000
Odd rows (1,3,5) are reconstructions, even rows (2,4,6) are the original data.

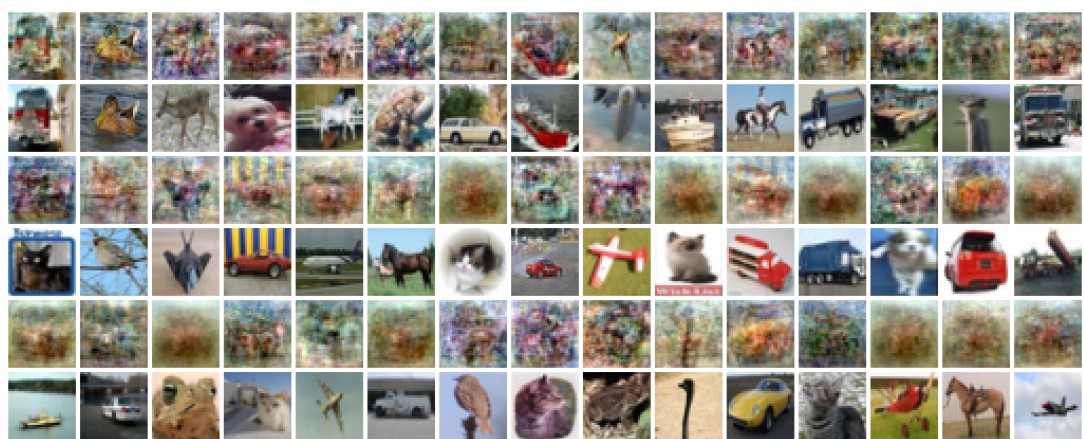

Figure 8: **Architecture:** $d$-100-100-1, **Samples:** 500
Odd rows (1,3,5) are reconstructions, even rows (2,4,6) are the original data.

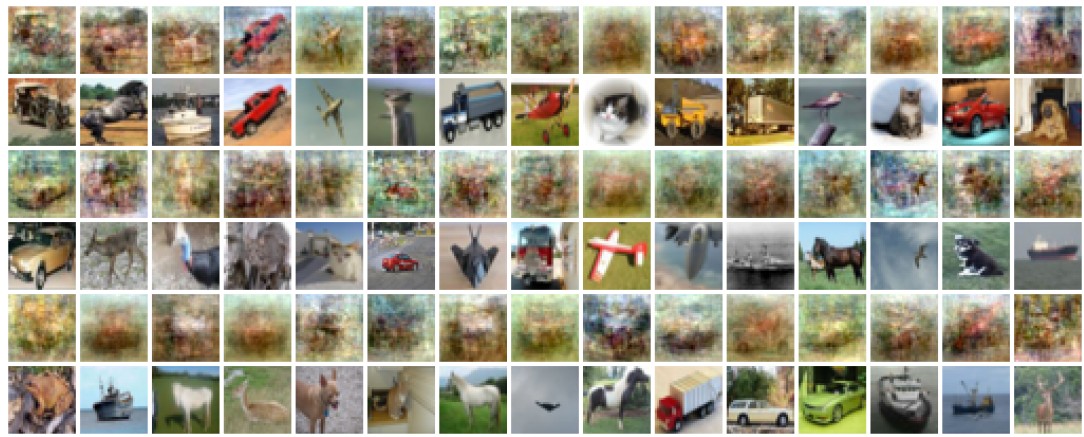

Figure 9: **Architecture:** $d$-1000-500-100-1, **Samples:** 500
Odd rows (1,3,5) are reconstructions, even rows (2,4,6) are the original data.

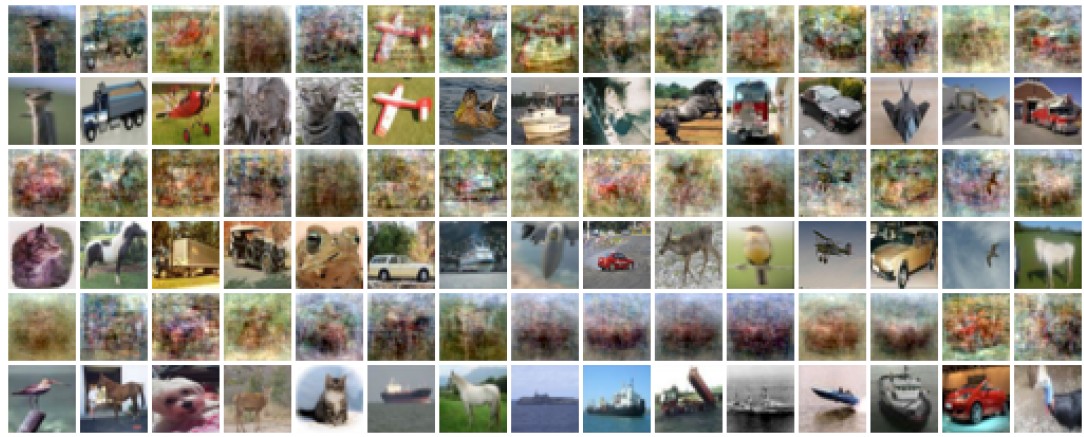

Figure 10: **Architecture:** $d$-1000-500-100-50-1, **Samples:** 500
Odd rows (1,3,5) are reconstructions, even rows (2,4,6) are the original data.

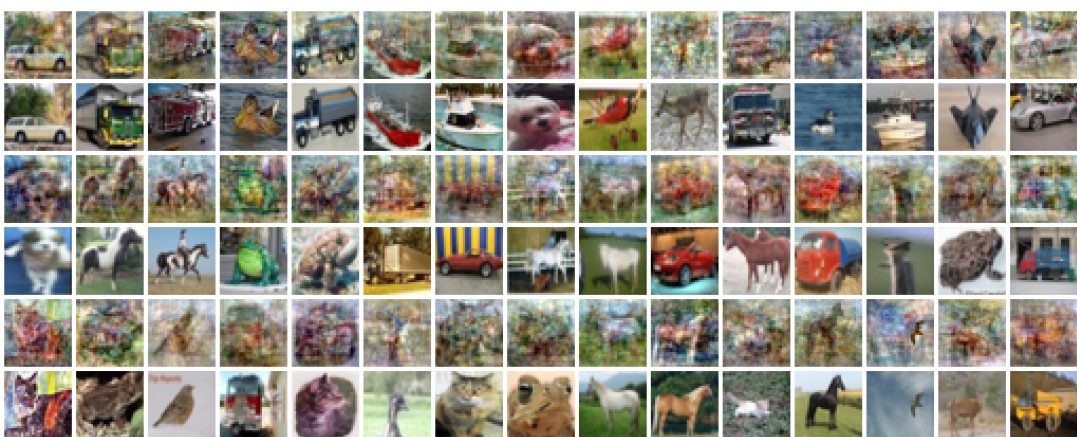

Figure 11: **Architecture:** $d$-1000-1000-1 (non-homogeneous), **Samples:** 500
Odd rows (1,3,5) are reconstructions, even rows (2,4,6) are the original data.

| Architecture | Training Set Size ($n$) | Train Loss | Test Accuracy | Test Loss |
|---|---|---|---|---|
| 1000-1000 | 50 | $1.5 \cdot 10^{-6}$ | 72% | 2.14 |
| 1000-1000 | 100 | $2.0 \cdot 10^{-6}$ | 74% | 2.41 |
| 1000-1000 | 500 | $4.0 \cdot 10^{-6}$ | 78% | 2.09 |
| 1000-1000 | 1000 | $5.5 \cdot 10^{-6}$ | 79% | 1.96 |
| 100-100 | 500 | $3.0 \cdot 10^{-7}$ | 77% | 2.72 |
| 1000-500-100 | 500 | $1.2 \cdot 10^{-7}$ | 78% | 3.14 |
| 1000-500-100-50 | 500 | $8.4 \cdot 10^{-7}$ | 77% | 2.57 |
| 1000-1000 (Non-Homogeneous) | 500 | $4.3 \cdot 10^{-6}$ | 77% | 2.12 |

Table 1: Train/Test loss and Test Error for models shown in Figure 4

In Figure 14 and Figure 15 we show all the weights, as images, of the first fully-connected layer of models trained on CIFAR10 and MNIST respectively. These are the same models as in Figure 3, i.e., there are 1000 weights. Some weights are indicative of several input samples, e.g., a plane from CIFAR10 and the digits 8 and 5 from MNIST. We note that our reconstruction scheme is able to reconstruct much more samples, and in better quality than is represented in these weights.

## C.3   Stretching the Theoretical Limitations

In this section we show results from several experiments which go beyond the theoretical limitations of Theorem 3.1.

| Experiment | Training Set Size ($n$) | Train Loss | Test Accuracy | Test Loss |
|---|---|---|---|---|
| Standard Initialization | 10 | $8.3 \cdot 10^{-7}$ | 71% | 1.68 |
| Standard Initialization | 50 | $1.5 \cdot 10^{-6}$ | 74% | 1.72 |
| SGD | 500 | $4.0 \cdot 10^{-6}$ | 77% | 2.21 |
| 10k Epochs (CIFAR10) | 500 | 0.0039 | 77% | 1.22 |
| 10k Epochs (MNIST) | 500 | 0.014 | 87% | 0.55 |

Table 2: Train/Test loss and Test Error for models shown in Figure 16

### C.3.1   Standard Initialization Scale

In this subsection we consider networks trained with standard initialization scales. We recall that in the experiments presented in Section 5 the first fully-connected layer is initialized to a Gaussian distribution with mean 0 and standard deviation $10^{-4}$, while the other layers are initialized by standard Kaiming initialization [He et al., 2015]. In Figure 17 and Figure 18 we show reconstructions of a model trained on CIFAR10 on 10 and 50 samples respectively, where the all the layers of the model are initialized by standard Kaiming initialization. The architecture of the model is $d$-1000-1000-1. We note that although the quality of the reconstructions is lower than when initializing the first layer with a small scale, there is still a strong signal that some of reconstructions correlate with training samples. It is an interesting future direction to improve the reconstruction quality for models with standard initialization.

In Figure 16 (a,b) we plot the SSIM score of each training sample against the output of the model. Note that indeed in these experiments the best SSIM score is lower than from other experiments presented in Figure 4. This corresponds to the lower quality of reconstructions when using standard initialization.

### C.3.2   Less Epochs

In the experiments from Section 5 we trained each model for $10^6$ epochs. The reason for this long training time is that Theorem 3.1 gives guarantees only when converging to KKT point. Such a convergence happens only after training until infinity, and longer training time may converge closer to the KKT point. In this section we provide reconstruction results for models trained for only $10^4$ epochs. Figure 19 and Figure 20 show reconstructions for models trained on 500 samples from

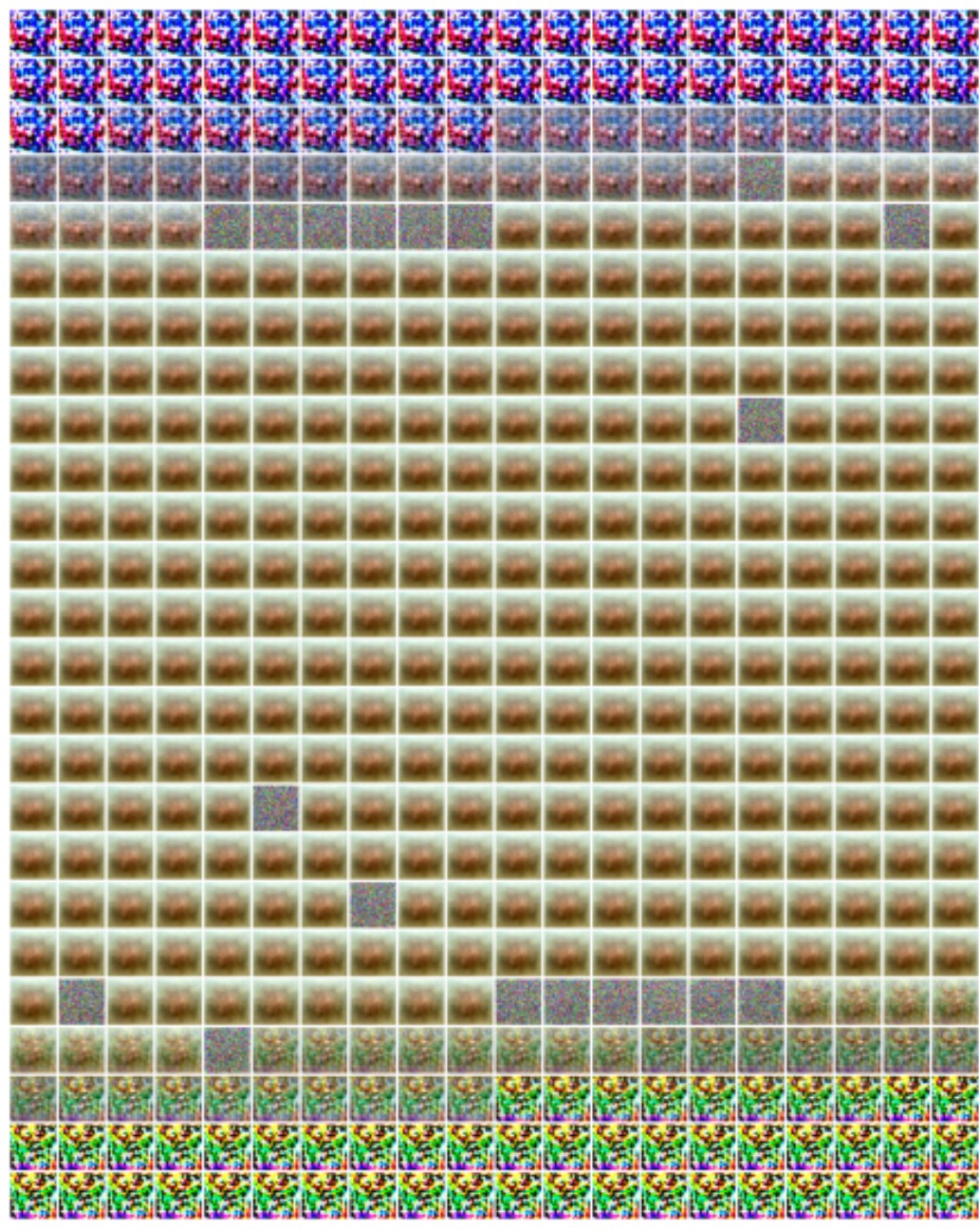

Figure 12: Model inversion attack on a model trained on CIFAR10, with 500 samples. We reconstructed a total of $40,000$ images using different initializations and hyperparameters. We sorted the results according to the model's output, and selected 500 representative with index $i = 0, 80, 160, ..., 40000$.

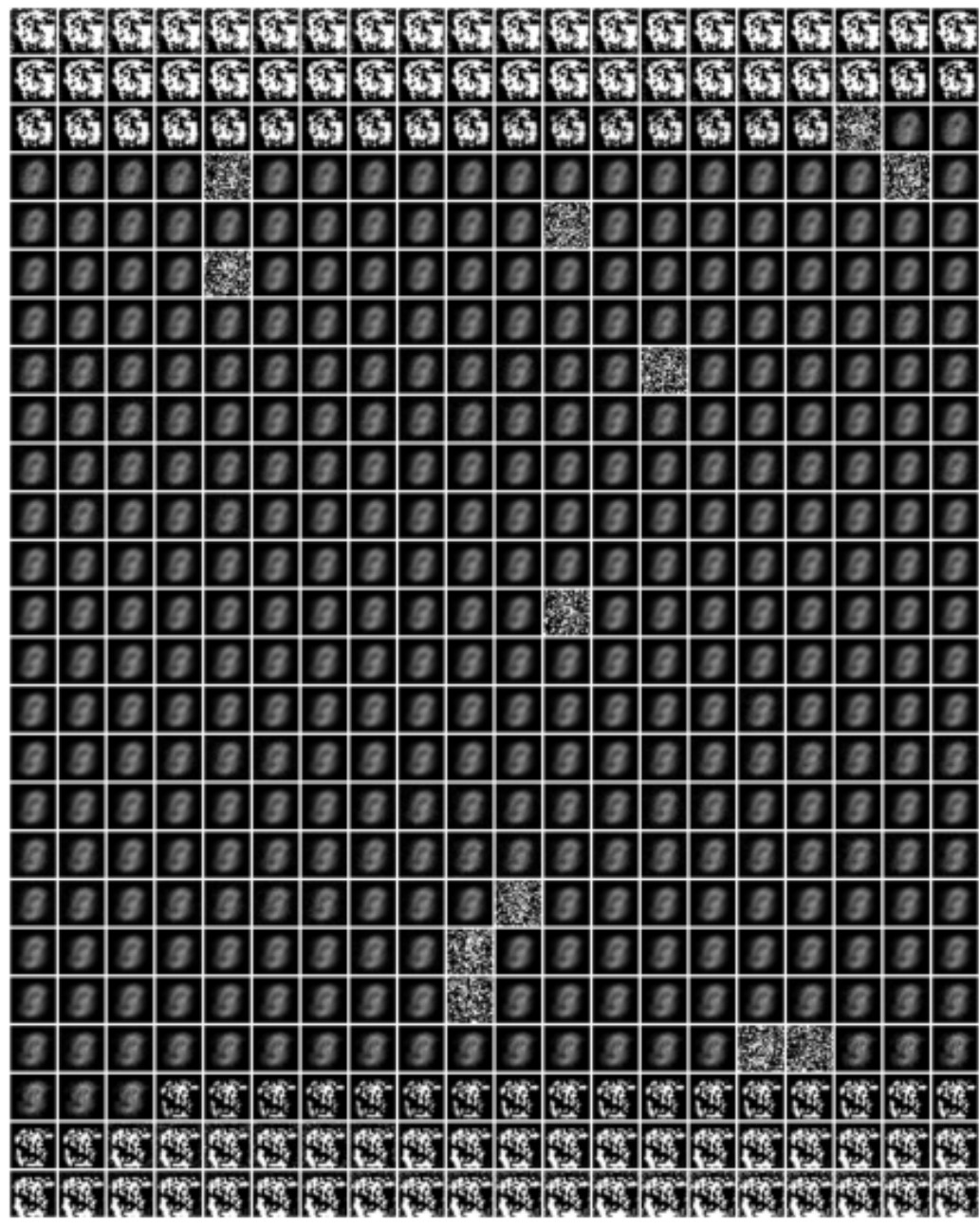

Figure 13: Model inversion attack on a model trained on MNIST, with 500 samples. We reconstructed a total of $40,000$ images using different initializations and hyperparameters. We sorted the results according to the model's output, and selected 500 representative with index $i = 0, 80, 160, ..., 40000$.

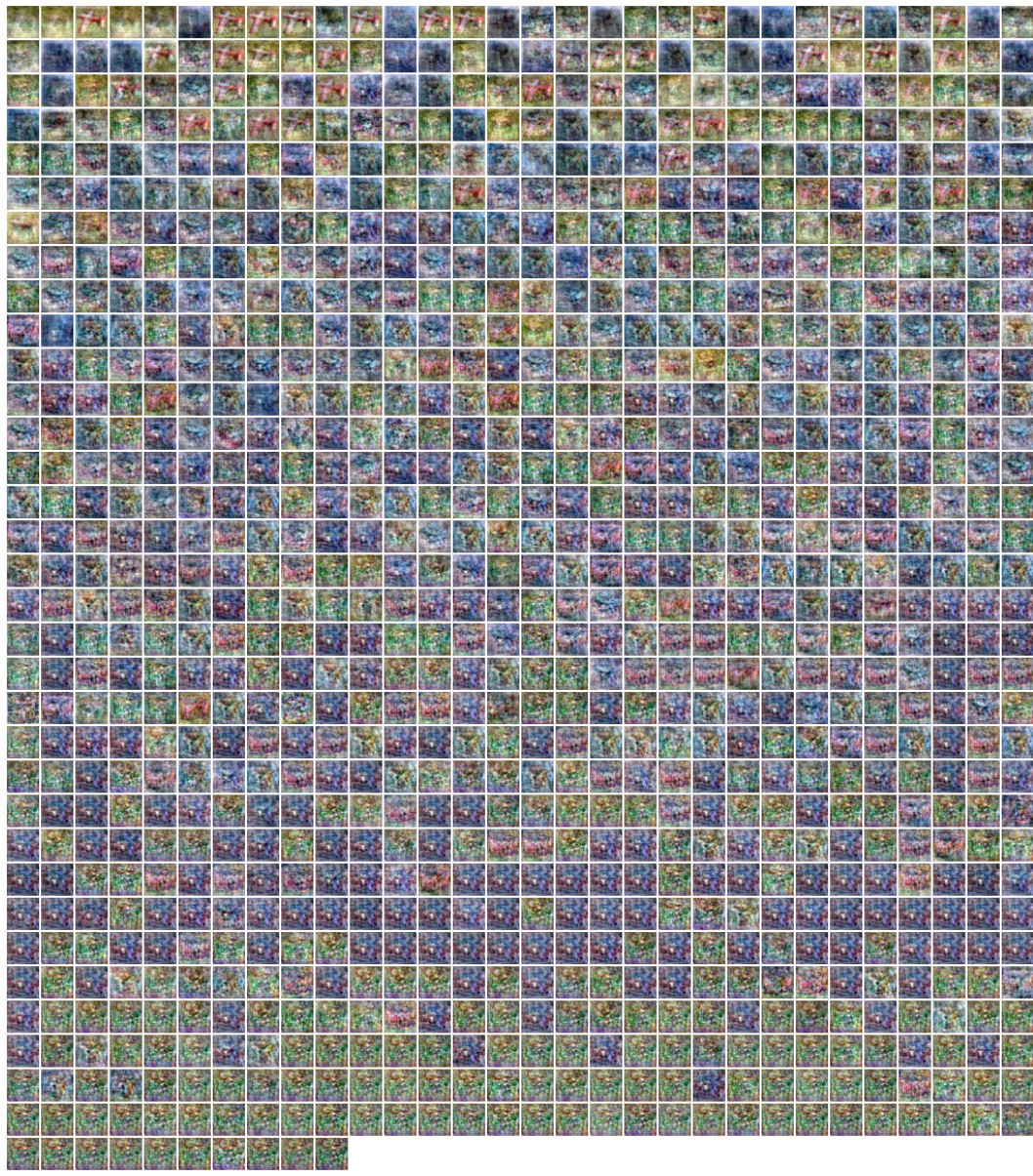

Figure 14: All the 1000 weights, shown as images, of the first fully-connected layer of a model trained on 500 samples on CIFAR10.

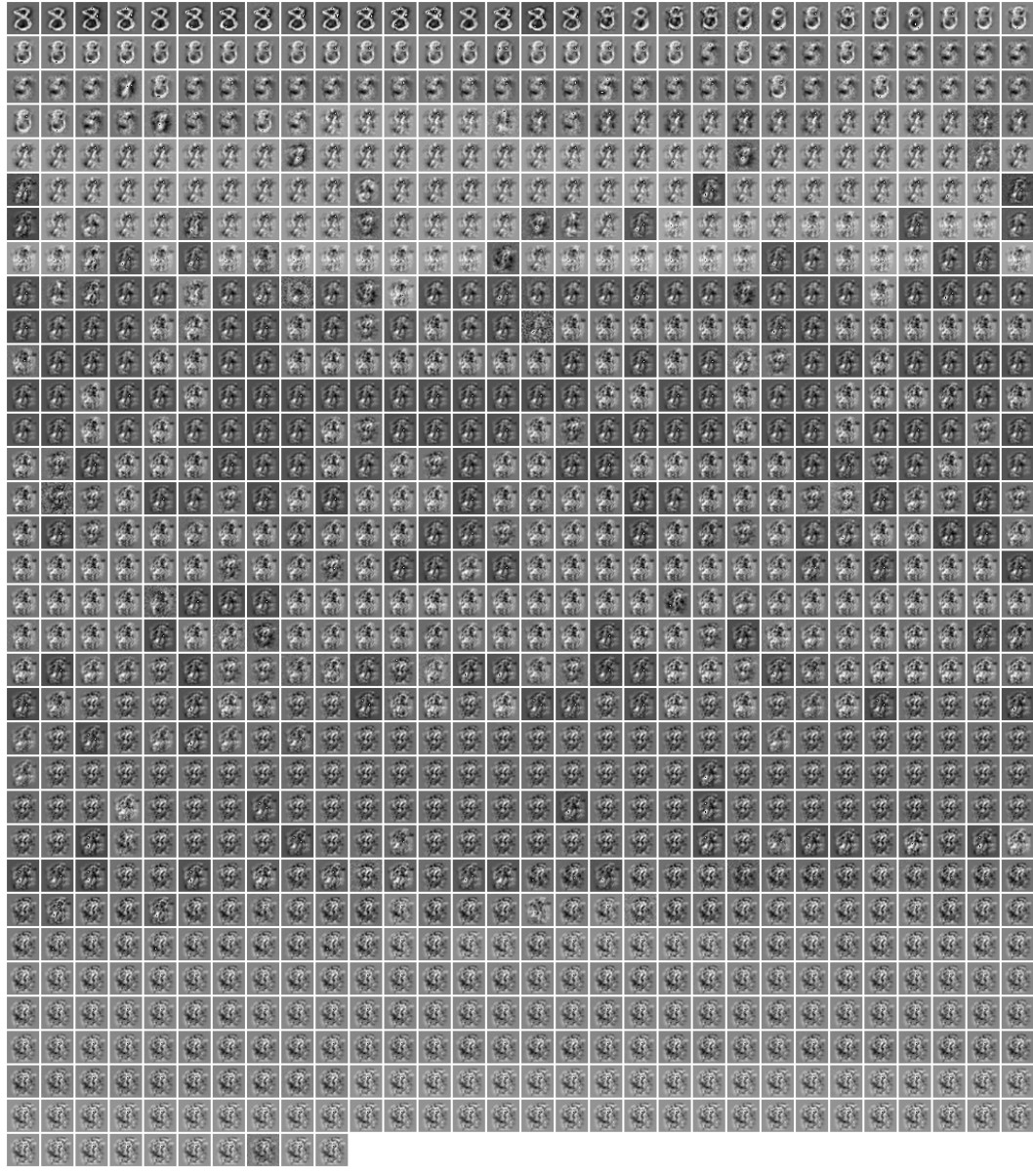

Figure 15: All the 1000 weights, shown as images, of the first fully-connected layer of a model trained on 500 samples on MNIST.

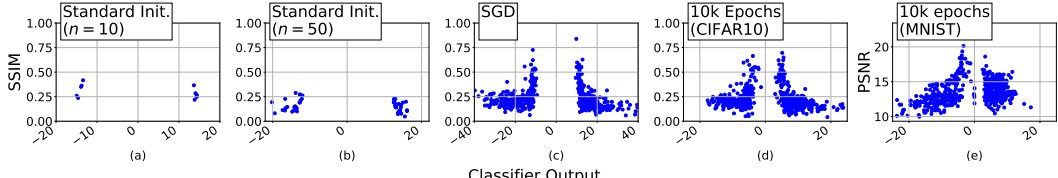

Figure 16: Each point represents a training sample. The y-axis is the highest SSIM score achieved by a reconstruction of this sample, the x-axis is the output of the model. From left to right: (a,b) models trained with standard Kaiming initialization in all layers on 10 and 50 CIFAR samples. (c) A model trained using SGD with a batch size of 50. (d,e) Models trained for $10^4$ epochs on 500 samples from CIFAR and MNIST respectively.

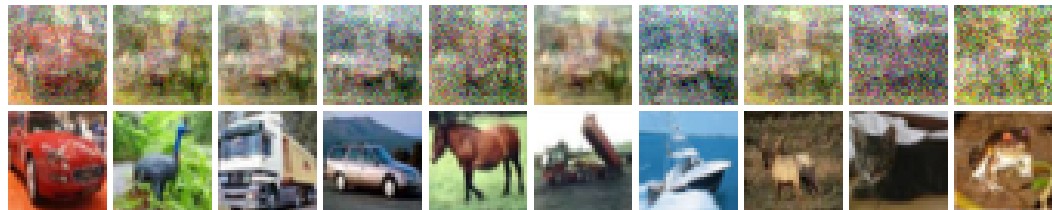

Figure 17: Reconstructions from a model trained on 10 CIFAR10 images with labels animals vs. vehicles. In the first row are the reconstructions, and in the second row are their corresponding nearest neighbor from the dataset (sorted by SSIM score).

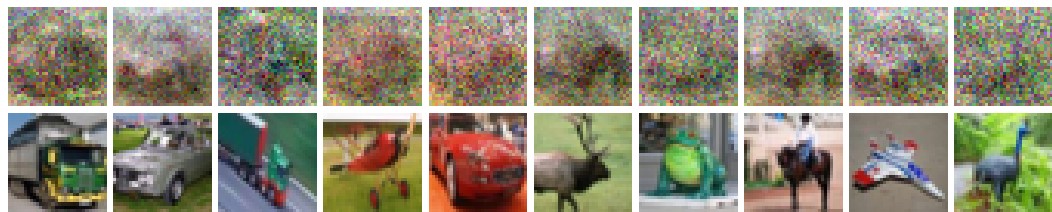

Figure 18: Top 10 reconstructions from a model trained on 50 CIFAR10 images with labels animals vs. vehicles. Top row shows reconstructions, and bottom row shows their corresponding nearest neighbor.

CIFAR10 and MNIST datasets respectively, with an architecture of $d$-1000-1000-1. It is clear that the quality of the reconstruction is very similar to when training for more epochs, this may indicate that even after significantly less training epochs the model converge sufficiently close to a KKT point.

In Figure 16 (d,e) we plot the SSIM score of each training sample against the output of the model. We note that we are able to reconstruct samples which appear approximately on the margin for both MNIST and CIFAR. In addition, the model for MNIST did not achieve 0 train error, and the margin is still very small. With that said, we are still able to reconstruct a large portion of the data with high quality. This goes beyond our theoretical limitations which have guarantees only for models which successfully label the entire training set.

### C.3.3 Mini-batch SGD

In the experiments from Section 5 we trained the models using full-batch gradient descent. This was done to align with the theoretical guarantees of Theorem 3.1, which assume training with gradient flow. In Figure 21 we show reconstructions from a model trained with mini-batch SGD, using a batch size of 50. The model is trained on 500 images from CIFAR10, and with an architecture of $d$-1000-1000-1.

In Figure 16 (c) we plot the SSIM score of each training sample against the output of the model. This plot shows that we indeed reconstruct samples that lie on the margin.

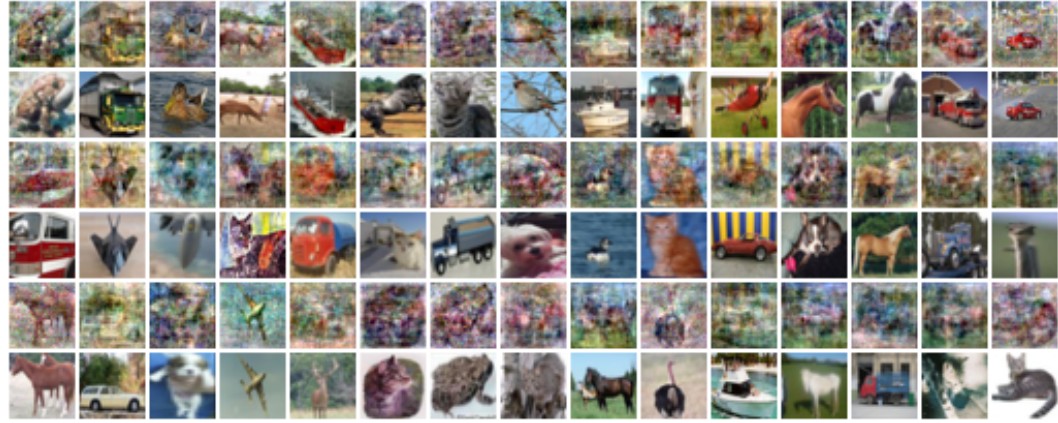

Figure 19: Reconstructions from a model trained for $10^4$ epochs on CIFAR10 with labels animals vs. vehicles. Odd rows (1,3,5) are reconstruction, and even rows (2,4,6) are their nearest neighbor from the training samples.

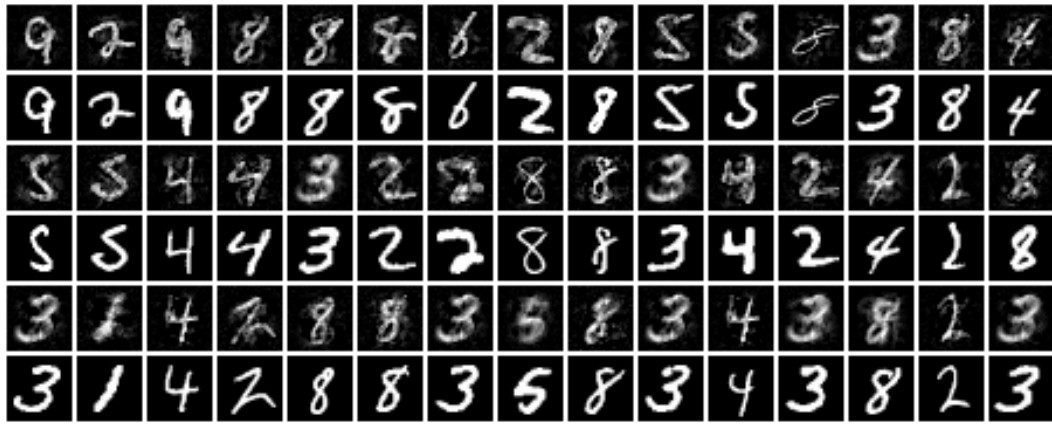

Figure 20: Reconstructions from a model trained for $10^4$ epochs on MNIST with labels odd vs. even. Odd rows (1,3,5) are reconstruction, and even rows (2,4,6) are their nearest neighbor from the training samples.

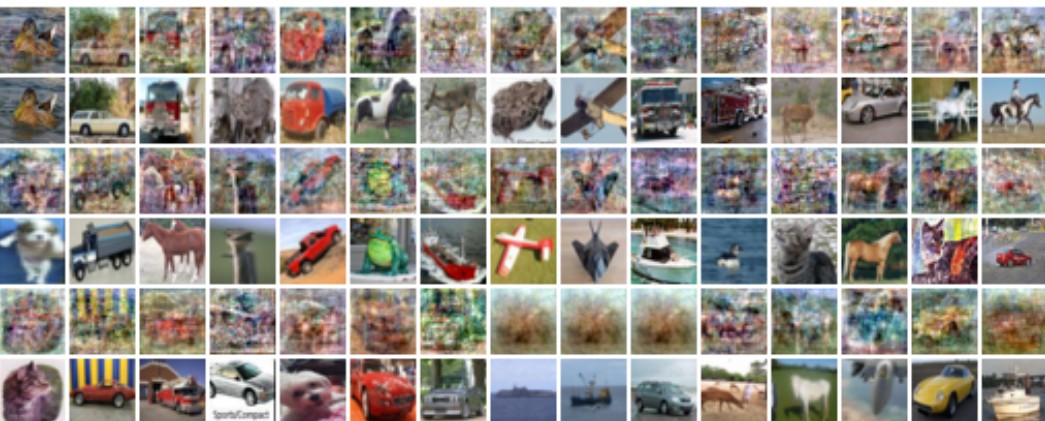

Figure 21: Reconstructions from a model trained using SGD with a batch size of 50. The model trained on 500 images from CIFAR10 with labels animals vs. vehicles. Odd rows (1,3,5) are reconstructions and even rows (2,4,6) are their nearest neighbor from the training dataset.