# OpenReview forum: "Reconstructing Training Data From Trained Neural Networks"
_NeurIPS.cc/2022/Conference — NeurIPS 2022 Accept_

### Official Review · Reviewer_feXv · 2022-07-08

**Rating:** 5
**Confidence:** 4
**Soundness:** 3 good
**Presentation:** 3 good
**Contribution:** 3 good

**Summary:**

The paper deals with an interesting problem. Reconstructing training data from weights. Mostly, it has been attempted to reconstruct data based on logit outputs only. The paper has far reaching impact on privacy. The paper proposes a method and demonstrates its effectivenss on layer MLPs.

**Questions:**

Could you add the needed evaluation (see above) or explain why the method works well in a practical setup?

**Limitations:**

I recommend that the author have a dedicated paragraph or subsection named "Limitations", where they discuss them.

**Strengths And Weaknesses:**

+ The problem is highly relevant.
+ Showing that training data can be accurately reconstructed from weights would be ground breaking and have far reaching practical implications with respect to privacy.
+ The optimization procedure is interesting with some nice tweeks.
- Unfortunately, and this is a key aspect, the paper falls short in evaluation. Evaluating on a practical network and a practical training setup (in terms of amount of data) is essential for the method to have far reaching impact and also be convincing. Since for simple architectures as those used in the paper trivial reconstructions (e.g. just visualizing weights) give already good results, the additional value of the method seems limited and it is also not clear whether it works at all in a practical setup - see detailed comments.

Detailed comments:
- Evaluation should be on a reasonable architecture: not just a two layer MLP, e.g., a CNN like ResNet or VGG. The reason why I am saying this, is for linear networks, it is known that weights might resemble class averages/ prototypes - the authors state this themselves (Figure 5b) - another reference showing this in a different context is, e.g., https://arxiv.org/abs/2202.08299. So it appears that 2 layer MLPs encode samples / class prototypes. The authors somewhat confirm this in Figure 5b, though admittedly their reconstructions are clearly superior to showing weights only. However, it is not clear, whether the approach would work on deep networks, i.e., is the optimization only good for fine-tuning already good prototypes? Would it fail (completely) if they did not exist? Where failure could be that only a very small fraction (I am talking 1/10^6 or so) of reconstructed images are actual training data. The authors themselves state that not only optimization attempts are successful and given that classifiers are subject to adversarial samples, the confidence in the generalizability of the findings to more realistic networks is very low.
- Evaluation should use a larger training dataset: The authors use only 500 samples. If they used just one sample per class, it would be quite likely that it resembles a class-average, i.e. that sample. It is not obvious whether their method works if trained on 5000 or 50000 samples.
- "not an actual examples" -> not actual samples
- we use the original test sets of MNIST/CIFAR10 with 10000/8000 images respectively, and labeled accordingl -> Cifar10 has 10000 test samples, does not it?

---

> ### Author Response · Authors · 2022-08-02
> **Rebuttal**
>
> We thank the reviewer for the feedback, but would like to point to a few factual errors in the review:
>
> 1) “The paper proposes a method and demonstrates its effectivenss on 2 layer MLPs”:
> In our experiments we use up to 5 layer MLPs, please see Figure 4.
>
> 2) “trivial reconstructions (e.g. just visualizing weights) give already good results” :
> We devote Section 5.4 and Appendix C.2 specifically to show that this is *not* the case. In particular, we visualize all the weights of the first layer (in Figures 14 and 15, as well as pages 23-24 in the Appendix), and it is clear from those figures and from Figure 5 that this method has far worse reconstruction performance than ours.
>
> 3) “The authors use only 500 samples. If they used just one sample per class, it would be quite likely that it resembles a class-average, i.e. that sample.”:
> First, we train up to 1000 samples, not 500 (see Figure 4 top right). Second, in our settings there are many samples per class, therefore reconstructing class averages would not be meaningful. One sample per class, as suggested by the reviewer, is a different and possibly much simpler setting than what is shown in our experiments.
>
> Regarding the CIFAR-10 test set, indeed it contains 10,000 samples. However, we consider binary classification of vehicles vs. animals (see line 193). Since the vehicles class contains only 4/10 of the total classes in CIFAR-10, there are only 4000 samples from the original test set for the vehicles class. Since we evaluate using a balanced test set, we only use 4000 samples for the animals class (even though there are a total of 6000 images in the original CIFAR-10 test set for this class). This sums to a total of 8000 samples in our test set.

---

> > ### Comment · Reviewer_feXv · 2022-08-08
> > **...**
> >
> > Thanks for your elaboration. I updated my rating after reading your response and the other reviewers. I am still sceptical as layed out in the initial review, and the response was only partially helpful to eliviate the concerns.

---

### Official Review · Reviewer_fWe7 · 2022-07-09

**Rating:** 6
**Confidence:** 4
**Soundness:** 3 good
**Presentation:** 3 good
**Contribution:** 3 good

**Summary:**

This paper gives a principled and novel scheme for reconstructing training set image only use parameters of a network. The approach is elegant and principled and the results are convincing.

**Questions:**

1. Being curious, have the authors considered other network architectures? (well I feel it is natural to consider them, and maybe the author have already tried the algorithm on these networks?) If not, why?

2. If a network is trained with differential privacy (there are plenty work study differentially private neural network training), how well would this approach work? Note that, it seems to me that, successful reconstruction seems to be in contradiction with DP

**Limitations:**

Potential negative societal impact because the adversaries can reconstruct training data just from network parameters. But this is for good reasons because it means this phenomenon should receive attention and one should defense mechanisms

**Strengths And Weaknesses:**

Strengths: Principled and elegant approach, with convincing experimental results

Weakness: One thing I am wondering about is other more complicated network architectures, especially those that have been used in practice today. For example, how about we apply this to CNN? What will happen? Note that even CNN should be regarded as a "baby version".

So to this end, I do have some concerns about the experiments regarding effectiveness of the approach on state-of-the-art neural networks

---

> ### Author Response · Authors · 2022-08-02
> **Rebuttal**
>
> We thank the reviewer for the feedback and the comments.
>
> Regarding extension to CNNs: We emphasize that Theorem 3.1 works as-is on convolutional neural networks, hence our reconstruction scheme should (in theory) be able to work on them.
> In this paper we focused on fully-connected networks, and we agree that generalizing our results to CNNs is a very interesting future research direction.
>
> Regarding training with differential privacy guarantees: Training a model using a differentially private algorithm gives very strong privacy guarantees against any data reconstruction attack, hence it should protect from our reconstruction scheme. We note that such algorithms inject noise to the training (usually to the gradient updates) which hurts the performance of the model. In particular, Theorem 3.1 and the KKT conditions will no longer work under this regime, hence we cannot expect our scheme to work. We also note that the noise from using mini-batch SGD is not enough to prevent our reconstruction scheme, as shown in Appendix C.

---

### Official Review · Reviewer_2grn · 2022-07-11

**Rating:** 8
**Confidence:** 3
**Soundness:** 3 good
**Presentation:** 4 excellent
**Contribution:** 4 excellent

**Summary:**

This paper proposes a new method for reconstructing parts of the training data from a given trained homogeneous binary classification network without any further prior knowledge. It relies on a theoretical result by Lyu and Li as well as Ji and Telgarsky, who showed that the gradient flow of the training loss of such a network converges in direction to parameters that solve a maximum margin problem. The optimality condition to the latter is used to phrase an optimization problem for recovering training data on the margin. The resulting approach is tested in various contexts for a 2d toy example, a binary MNIST image classification, and a binary CIFAR-10 classification, and demonstrates significant abilities to recover some parts of the training data.

**Questions:**

- The training images to be reconstructed are initialized from a zero mean Gaussian distribution, i.e. roughly half of the values being negative besides an explicit regularization term the tries to enforce a [0,1] box constraint. This seems contradictory. Is it not possible to initialize with a uniform distribution in [0,1]^d?
- Constraints (such as lambda \geq 0 or the abovementioned box constraint) are implemented as penalties instead of running projected gradient descent (i.e. enforcing hard constraints explicitly). Is there a specific reason?
- To be very picky I suggest to write (5) as $\lambda_i (y_i \phi(\tilde{\theta};x_i)-1) = 0$ because strictly speaking I do not think $\lambda_i$ has to be greater than 0 if $y_i \phi(\tilde{\theta};x_i)=1$. And in line 159 I suggest not to use the word "converge" in "we converge to some point $\theta$" because the training setting is designed such that no minimizer exists (the norm of $\theta$ should slowly go to infinity and it can only converge in direction).

**Limitations:**

Yes, there are several limitations in terms of the investigated network architectures and a number of open research questions, but the authors have adequately addressed them.

**Strengths And Weaknesses:**

To my mind this is a really nice paper, because it exploited a theoretical result to propose a completely new method for reconstructing the data a network has been trained on, which yields significantly better results than competing model inversion schemes. I agree with the authors that a significant amount of training data can be reconstructed surprisingly well. While the investigated network architectures as well as the allowed settings (requiring all training examples to be classified correctly such that the loss converges to zero) remain limited, I still believe the results to be very significant as they lead to several further questions, including why the optimization problems works well at all, how many training data points typically are on the margin, and if training data points that are not on the margin are protected from possible recovery attacks as well as possible further characterizations for stochastic gradient descent, other training methods, multi-class classification network, or regression networks. It reveals interesting properties of the trained weights which is why I consider this to be a strong paper.

In terms of weaknesses, the paper is of course limited to a specific setting, which is, however, understandable considering it is the first work in this direction. Besides this, I only have some minor points that I am phrasing as questions below.

---

> ### Author Response · Authors · 2022-08-02
> **Rebuttal**
>
> We thank the reviewer for the comments and for the positive feedback. To answer your questions:
>
> 1) Thanks for pointing this out, this is in fact a typo. The inputs are represented in [-1,1] (and not [0,1] as is erroneously written), and the prior bounds are [-1,1] respectively. This will be fixed (both in lines 213-214 and line 280).
>
> 2) There is no particular reason, this penalty seemed to work well in practice. In addition to the suggestion made by the reviewer, it is also possible to optimize over $\lambda^2$ instead of $\lambda$, and thus enforce those variables to be non-negative.
>
> 3) This is indeed a typo and will be fixed, it should be $\lambda_i=0$ if $y_i\Phi(\tilde{\theta},x) \neq 1$. We will also change line 159 as the reviewer suggested.

---

> > ### Comment · Reviewer_2grn · 2022-08-09
> > **Keeping my score**
> >
> > I'd like to thank the authors for their answers and I have no further questions. While I understand the concerns of other reviewers, I'm weighting the conceptual contribution stronger than the practical limitations such that I am keeping my score.

---

### Official Review · Reviewer_gmuC · 2022-07-11

**Rating:** 7
**Confidence:** 4
**Soundness:** 4 excellent
**Presentation:** 4 excellent
**Contribution:** 3 good

**Summary:**

The authors in this paper address a challenging, and interesting problem in the current deep neural network literature. Though it is widely believed that neural networks essentially memorize, thereby overfit to training samples, it is usually not very clear how one might be able to achieve this. This paper proposes a relatively clean and straightforward formulation to compute the training data points. The authors rely on the results of [Ji and Telgarsky, 2020] to frame the problem as discussed in Equations (2) - (5). The technique hinges on the fact that at the training data points, a linear combination of the derivatives of the network parameters can predict the parameters themselves. A very interesting observation in my opinion.


**Questions:**


I wonder if the authors have suggestions, for a possible way to extend the current work to networks with skip connections ?


**Ethics Review Area:**

["I don’t know"]

**Limitations:**

Not general enough to be applicable to Resnet networks.

**Strengths And Weaknesses:**

Strengths :

1. Clean formulation of the problem in terms of  reconstruction loss to find the training points.
2. The approach produces convincing results in some of the standard computer vision datasets like MNIST and CIFAR10.

Weakness :

1. This work cannot be extended to CNNs with skip connections  easily due to the homogeneity assumption. Which limits the test error on CIFAR10 due to the above limitation is 71%.

---

> ### Author Response · Authors · 2022-08-02
> **Rebuttal**
>
> We thank the reviewer for the thorough review and the comments.
>
> Extension to CNNs: We emphasize that Theorem 3.1 works as-is on convolutional neural networks, hence our reconstruction scheme should (in theory) be able to work on them.
> In this paper we focus on fully-connected networks. Generalizing our results to CNNs is a very interesting future research direction.
>
> Extension to skip connections: We note that although the theory is limited to homogeneous networks, in practice our reconstruction scheme may work on non-homogeneous networks as well. In particular, Figure 4 bottom right shows that we are able to reconstruct CIFAR images even for a non-homogeneous network. We hope that extending our scheme to skip connections should also be possible.

---

### Meta-Review · Area_Chair_rEW3 · 2022-08-23

**Recommendation:** Accept
**Confidence:** Certain

**Metareview:**

This paper proposed a new algorithm to reconstruct a subset of training examples from a trained homogeneous binary classification neural network. Although there are still some limitations such as the zero training loss and homogeneity assumption, as well as limited experiments beyond MLPs, the reviewers also acknowledge that this results is very interesting and reveals an important property of deep neural networks that could potentially have far-reaching implications for privacy and security.

**Award:**

No

---

### Decision · Program_Chairs · 2022-09-14

Accept